# Identification of Human UDP-Glucuronosyltransferase Involved in Gypensapogenin C Glucuronidation and Species Differences

**DOI:** 10.3390/ijms24021454

**Published:** 2023-01-11

**Authors:** Juan Chen, Lin Qin, Xingdong Wu, Daopeng Tan, Yanliu Lu, Yimei Du, Di Wu, Yuqi He

**Affiliations:** 1Guizhou Engineering Research Center of Industrial Key-Technology for Dendrobium Nobile, Zunyi Medical University, Zunyi 563000, China; 2Joint International Research Laboratory of Ethnomedicine of Ministry of Education, Zunyi Medical University, Zunyi 563000, China

**Keywords:** gypensapogenin C, glucuronidation, human liver microsomes, UGT1A4, species differences

## Abstract

Gypensapogenin C (GPC) is one of the important aglycones of *Gynostemma pentaphyllum* (GP), which is structurally glucuronidated and is highly likely to bind to UGT enzymes in vivo. Due to the important role of glucuronidation in the metabolism of GPC, the UDP-glucuronosyltransferase metabolic pathway of GPC in human and other species’ liver microsomes is investigated in this study. In the present study, metabolites were detected using high-performance liquid chromatography–tandem mass spectrometry (LC–MS/MS). The results show that GPC could generate a metabolite through glucuronidation in the human liver microsomes (HLMs). Additionally, chemical inhibitors combined with recombinant human UGT enzymes clarified that UGT1A4 is the primary metabolic enzyme for GPC glucuronidation in HLMs according to the kinetic analysis of the enzyme. Metabolic differential analysis in seven other species indicated that rats exhibited the most similar metabolic rate to that of humans. In conclusion, UGT1A4 is a major enzyme responsible for the glucuronidation of GPC in HLMs, and rats may be an appropriate animal model to evaluate the GPC metabolism.

## 1. Introduction

*Gynostemma pentaphyllum* (Thunb.) Makino (GP) originated in South China and is a herbaceous climbing vine of the genus *Gynostemma* in the family Cucurbitaceae, now widely distributed in China, Japan, Myanmar, and India [1,2]. In the 2022 edition of the Complete Catalogue of Medicines and Edibles, *Gynostemma* is listed as a medicinal and food-related health product. As an edible medicinal plant, GP has now been developed into a variety of products and preparations, including teas, beverages, capsules, tablets, and oral solutions [3]. According to phytochemical studies, the main chemical constituents of GP are gypenosides (GPs), polysaccharides, flavonoids, phytosterols, amino acids, and vitamins [4,5,6,7]. Up to now, 328 GPs [3], 10 flavonoids, 7 flavonoid glycosides [8], 18 amino acids, 23 inorganic elements, and 19 microelements [9] have been isolated from GP. Previous studies found that the in vivo antitumor assays of GP showed stronger antitumor activity than those in vitro, suggesting that the metabolites of GP in vivo may exert greater antitumor effects than those of the prototype GP [10]. Therefore, it is necessary to study the metabolic process of GP.

GPs are the main components of GP and their main pharmacologically active ingredients. The clinical efficacy of GPs has been studied in a wide range of experiments, including anticancer [11], hypolipidemic [12], hypoglycemic [13], anti-inflammatory [11], antioxidant, and hepatoprotective [14]. However, most of the studies on GPs are focused on their pharmacological activity, and metabolic studies have not yet been conducted. GPs are a class of dammarane-type tetracyclic triterpenoid saponins that contain a large number of glycosyl structures, mostly at C-3 and C-20. Through literature research, we found that GPs still have the characteristics of glycosylation after hydrolysis to aglycones, which is very likely to combine with glycosyl again.

Phase II metabolism is the further binding of the drug itself or its Phase I metabolites to endogenous substances in the body to increase their water solubility and facilitate elimination from the body, mainly including glucuronidation, sulfation, methylation, acetylation, and binding to glutathione [15], in which UGT enzyme-mediated glucuronidation metabolism is involved in approximately 40–70% of clinical drug metabolism and accounts for of more than 35% [16,17]. Although Phase I metabolizing enzymes are involved in the metabolism of most compounds, Phase II metabolizing enzymes, especially UGT enzymes, play a more prominent role in drug detoxification and drug clearance [18]. Previous studies showed that the Phase I metabolism of GPs is mainly through the epoxidation, cyclization, deglycosylation, and dehydration of double bonds [19,20]. In addition, GPs are inhibitors of human cytochrome P450 (CYP) enzymes and have the strongest inhibitory effect on CYP2D6 (IC_50_ value of 1.61 μg/mL) [21]. For Phase II metabolism, Zhuo et al. [22] found that GPs could induce the expression of drug-metabolizing enzymes such as UGT and GST in the liver tissue of rats, and increase the activity of drug-metabolizing enzymes. However, specific studies on the Phase II metabolism of GPs have not been reported. Gypensapogenin C (GPC) is one of the main secondary products obtained from the acid hydrolysis of GPs (structure shown in Figure 1) and structurally belongs to dammarane-type triterpene aglycones. Therefore, GPC has greater potential to bind to UGT enzymes in vivo and is more likely to undergo glucuronide-binding reactions compared to other metabolic enzymes.

On the basis of the above literature research, our study aims to investigate the glucuronidation metabolic pathway of GPC. Metabolic enzymes involved in GPC glucuronidation in human liver microsomes (HLMs) by chemical inhibitors and recombinant human UGT enzymes are identified. Differences in metabolic species can also identify suitable animal models for the experimental studies of GP. In general, the present study is the first to investigate the glucuronidation metabolism of aglycone, the active monomers of GP.

## 2. Results

### 2.1. Glucuronidation of GPC in Human Liver Microsomes

To obtain more information on the glucuronidation products of GPC in HLMs, this study scanned the products using positive and negative ions. The results are shown in Figure 2A. One new chromatographic peak appeared in the GPC incubation group (red) under negative ion mode, with a retention time of 4.07 min. Further analysis showed that this peak was not observed in the blank control group (black) and the negative control group (blue). The peak was UGPGA and liver microsome-dependent. Therefore, we speculate that it may be a product of GPC glucuronidation.

Further mass spectrometric analysis of the metabolite indicated the presence of metabolite ions at *m*/*z* 613.2, which differed from GPC at *m*/*z* 437.6 by 176.0 Da, exactly the molecular weight of glucuronide. A secondary mass spectrometry scan was performed on the excimer ion ([M − H]^−^) of this metabolite. The MS/MS plot appeared to generate fragment ion 437.6 of the metabolite stripped of 176.0 Da (Figure 2B), which just corresponded to the GPC portion of the glucuronide conjugate. The 113.1 was formed by the further breakage of *m*/*z* 175.0. The results were exactly in line with the predicted glucuronidation conjugate of GPC, indicating that HLMs could catalyze the glucuronidation reaction of GPC to produce a conjugate product.

### 2.2. Kinetics of GPC Glucuronidation in HLMs

To further assess the glucuronidation metabolic activity of GPC in HLMs, our study determined the enzymatic kinetics of GPC in HLMs and its parameters in the range of substrate concentrations of 2–200 μM. First, the Eadie–Hofstee plot showed that the glucuronidation of GPC in HLMs was consistent with substrate inhibition kinetics (Figure 3). The metabolic kinetic parameters were obtained by the nonlinear fitting of the kinetic data with GraphPad Prism 8.0 software. As shown in Table 1, the *K_m_* value of GPC metabolism via the glucuronidation in HLMs was 15.36 ± 5.26 μM, *V_max_* was 0.42 ± 0.09 nmol/min/mg protein, and the inhibition constant *K_i_* of 42.68 ± 13.63 μM, and *CL_int_* of 27 μL/min/mg protein.

### 2.3. Chemical Inhibition Experiment

The inhibition of GPC (15 μM) glucuronidation metabolism was studied using four concentration levels (1, 10, 50, and 200 μM) of chemical inhibitors hecogenin, phenylbutazone, fluconazole, and magnolol [23]. The results are shown in Figure 4. Neither fluconazole nor phenylbutazone inhibited the glucuronidation activity of GPC, which indicated that UGT1A1, 1A3, UGT1A6, 1A7, 1A8, 1A9, 1A10, 2B15, and 2B7 were not involved in the glucuronidation metabolism of GPC. Magnolol showed an insignificant inhibition of GPC metabolism at low concentrations (1–50 μM). Its inhibitory effect was also only 50% at high concentration (200 μM), which showed that UGT1A9 was not the main enzyme that metabolized GPC. However, hecogenin had the most pronounced inhibitory effect on the glucuronidation metabolism of GPC. The inhibitory effect of hecogenin on GPC glucuronidation reached more than 40% at low concentration (1 μM) and became stronger at a higher concentration (70% inhibition of GPC glucuronidation metabolism at 10 μM), indicating that UGT1A4 might dominate the GPC glucuronidation metabolism process.

### 2.4. Glucuronidation Metabolism of GPC in Recombinant Human UGT Enzymes

Ten recombinant human UGT enzymes (UGT1A1, 1A3, 1A4, 1A6, 1A7, 1A8, 1A9, 1A10, 2B7, and 2B17) were used to catalyze GPC glucuronidation and validate the primary metabolic enzymes of GPC. The results are shown in Figure 5. UGT1A4 had the highest catalytic activity at GPC concentrations of 1 and 10 μM, while none of the remaining UGT isoenzymes showed activity. The results were consistent with those of chemical inhibitor studies, indicating that UGT1A4 was the isoenzyme involved in GPC glucuronidation in HLMs.

### 2.5. Glucuronidation Metabolism of GPC among Different Species

To further observe the difference in the GPC glucuronidation metabolism, the research characterized the metabolic profile of GPC in the liver microsomes of different species. As shown in Figure 6, GPC generated a glucuronidation product in the liver microsomes of rat, mice, dogs, rabbits, bovines, pigs, and monkeys. The metabolite was UDPGA-dependent and microsome-dependent, and had the same retention time and molecular weight as the product in HLMs, Therefore, it could be identified as a glucuronidation product of GPC, indicating that GPC could be metabolized by glucuronidation in the liver of humans, rats, mice, dogs, rabbits, bovines, pigs, and monkeys.

In addition, a semiquantitative comparison of the peak area ratios of GPC glucuronidation product generation at two concentrations of 10 and 50 μM (Figure 7) showed that the product was generated in the highest amounts in the microsomal incubation system of dogs, followed by rabbits and mice. Trace amounts of the product were also generated in humans, rats, bovines, pigs, and monkeys, indicating that the liver of these five species has a relatively weak capacity to metabolize GPC glucuronidation.

### 2.6. Enzymatic Kinetic Analysis of GPC in Liver Microsomes of Different Species

On the basis of the above results, to characterize its metabolic profile and the rate of the glucuronidation metabolism, our work further performed enzymatic kinetic studies. The liver-microsome-mediated glucuronidation of GPC in eight species conformed to the substrate inhibition equation (Figure 8). The enzymatic kinetic parameters were also measured separately, and the results are shown in Table 1. For *K_m_*, dog (9.56 μM) = pig (9.58 μM) > mouse (10.04 μM) > bovine (12.24 μM) > human (15.36 μM) > rabbit (16.19 μM) > rat (94.07 μM) > monkey (107.10 μM), indicating that the affinity of UGT metabolizing enzymes to GPC in dogs, pigs, mice, bovines, humans, and rabbits was much higher than those of rats and monkeys, among which the affinity of dogs and pigs was the highest.

To assess the ability of these eight species to metabolize GPC via the glucuronidation pathway, we calculated the intrinsic clearance (*CL_int_*) of various species to metabolize GPC glucuronidation by *V_max_*/*K_m_*. For *CL_int_*, dog (2121 μL/min/mg) > mouse (690 μL/min/mg) > rabbit (304 μL/min/mg) > bovine (259 μL/min/mg) > pig (147 μL/min/mg) > monkey (77 μL/min/mg) > rat (40 μL/min/mg) > human (27 μL/min/mg) (Table 1). The results showed that dogs had the highest clearance of GPC, followed by mice, rabbits, and bovines, while humans had the lowest clearance of GPC, which was consistent with the results of the species differences observed in Section 2.5 above. However, since rats had the most similar intrinsic clearance rate to that of humans, it could be used as the most suitable animal model for preclinical studies of GPC.

## 3. Discussion

Aglycones are the main active component of GP with a wide range of pharmacological activities such as antitumor, anti-inflammatory, antidiabetic, and neuroprotective [11,13] In contrast to extensive pharmacological activity studies, the metabolic pathways and metabolic behaviors regarding GP in humans and experimental animals have not been investigated. Our study found a new chromatographic peak in the HLM incubation system for the first time. Further LC–MS/MS verified that the molecular weight of the glucuronidation product differed by just 176.0 Da from the GPC, which exactly corresponds to the glucuronide molecule. Usually, glucuronidation catalyzed by UGT enzymes occurs mainly in compounds containing hydroxyl, amine, and carboxyl groups, and chiral carbon atoms in the chemical structure [18]. Li et al. [24] clarified the glucuronidation metabolic site of 20(S)-protopanaxadiol (PPD) as C-3-OH using NMR. GPC and PPD are structurally dammarane-type tetracyclic triterpenoids, both of which contain a hydroxyl group at the C-3 position. Therefore, we speculate that the glucuronidation type of GPC is *O*-glucuronidation, which may occur in the C-3 hydroxyl group.

The identification of enzyme families involved in drug metabolism is important for predicting drug-drug interactions and interindividual variability [25]. For chemical inhibitors, only two specific inhibitors of UGT enzymes have been identified, namely, hecogenin (a specific inhibitor of UGT1A4) and fluconazole (a specific inhibitor of UGT2B7). In addition, our study used phenylbutazone and magnolol for chemical inhibition studies. Phenylbutazone inhibited UGT1A1, 1A3, 1A6, 1A7, 1A9, 1A10, and 2B15 [26] with poor specificity. Magnolol selectively inhibits UGT1A9 [27]. In this study, the four chemical inhibitors above combined with recombinant human UGT enzyme were used to validate the glucuronidase of GPC. The results show that UGT1A4 was the primary isoform enzyme. UGT1A4 is a type of UGT1A isoform enzyme that specifically catalyzes the *N*-glucuronidation reaction and is mainly expressed in the human liver [28,29,30]. However, with the advancement of science, some scholars have also demonstrated that UGT1A4 is capable of catalyzing *O*-glucuronidation. Li et al. [24] identified the glucuronidation metabolizing enzyme of PPD and confirmed that UGT1A4 is the primary isoform enzyme involved in its glucuronidation metabolism and that the metabolic site is C-3-OH. Moreover, UGT1A4 is one of the major UGT enzymes that catalyze the anticancer drug apatinib-*O*-glucuronidation [31]. Our study also further demonstrates that UGT1A4 can catalyze *O*-glucuronidation, which is in agreement with previous studies.

Traditionally, drug metabolism is an important factor in determining drug pharmacokinetics, toxicity, and efficacy [32]. Therefore, it is necessary to select suitable animal models to assess drug metabolism studies in the preclinical phase. In our study, dog and minipig liver microsomes showed the highest affinity for GPC with *K_m_* values of 9.56 ± 2.83 μM (dog) and 9.58 ± 2.32 μM (minipig), respectively. MkLM had the lowest affinity for GPC with a *K_m_* value of 107.10 ± 73.18 μM. Furthermore, the highest metabolic differences were observed between dog and human. Regarding metabolic efficiency, it is not sufficient to evaluate the metabolic rate of the drug using only *V_max_* and *K_m_* values, so the metabolic efficiency of the drug was measured with the intrinsic clearance *CL_int_* (*V_max_*/*K_m_*) in the present study [33]. GPC had the highest glucuronidation activity in dogs, followed by mice, rabbits, and bovines, and the glucuronidation metabolic activity for GPC was much higher in these four species than that in humans, implying that using these four species as animal models instead of human for GP metabolic studies may not be appropriate. The species with similar intrinsic clearance to human *(CL_int_* = 27 μL/min/mg protein) is the rat (*CL_int_* = 40 μL/min/mg protein), indicating that rats have similar glucuronidation metabolic activity to that of humans. Therefore, the rat may be the most suitable animal model for evaluating GPC or GP in preclinical studies.

In general, GPC can be rapidly metabolized by human UGT enzyme and generates a glucuronide. Enzymatic kinetics, chemical inhibitors, and recombinant human UGT enzymes have shown that UGT1A4 plays an important role in the liver glucuronidation metabolism of GPC. Extensive glucuronidation can result in drugs with lower bioavailability [34,35]. Previous pharmacokinetic experiments on GPs have shown that GPs have poor intestinal absorption [20], which could also be related to the glucuronidation of GPC. Moreover, since there are many drugs metabolized by UGT1A4 enzymes in clinical practice, such as amitriptyline and trifluoperazine, the occurrence of drug–drug interactions should be noted when GPs are used in combination with these drugs metabolized by UGT1A4.

## 4. Materials and Methods

### 4.1. Chemicals and Reagents

Uridine 5′-diphosphoglucuronic acid trisodium salt (UDPGA) and tris base were purchased from Sigma-Aldrich (St. Louis, MO, USA). Alamethicin and magnesium chloride hexahydrate were purchased from Aladdin Chemicals (Shanghai, China). The 20(*S*)-protopanaxatriol (PPT) was purchased from Chengdu Desite Bio-Technology (Chengdu, China). Human (HLMs, livers from 10 human donors of mixed sex), male New Zealand rabbit (RaLM, Lot no. YLHY), male Sprague-Dawley rat (RLM, Lot no. WJYW), male ICR/CD-1 mouse (MLM, Lot no. ZJCZ), male Beagle dog (DLM, Lot no. TXJT), and male bovine (BLM, Lot no. ANSD) liver microsomes were purchased from the Research Institute for Liver Diseases Co., Ltd. (RILD, Shanghai, China). Male pigs (PLM, Lot no. M10031) and male monkeys (MkLM, Lot no. M10003) were purchased from Prime Tox (Wuhan, China). Recombinant human UGT1A1, 1A3, 1A4, 1A6, 1A7, 1A8, 1A9, 1A10, 2B7, and 2B17 were purchased from Cypex (Scotland, UK). Chemical inhibitors (hecogenin, phenylbutazone, fluconazole, magnolol) were purchased from Shanghai Standard Technology Co., Ltd. (Shanghai, China). All microsomes and UGT enzymes were stored at −80 °C. All other chemicals were of HPLC grade or the highest grade available on the market.

### 4.2. Separation and Purification of GPC

GPC was separated and purified from GPs by our group, and its structure is shown in Figure 1. The separation method is as follows [36]: The GPs were hydrolyzed with hydrochloric acid to obtain the total saponin of *Gynostemma pentaphyllum*. Then, it was separated by silica gel column chromatography to obtain the crude product of total saponin. The crude product was further finely separated using silica gel column chromatography, semipreparative high-performance liquid chromatography, and preparative high-performance liquid chromatography. Furthermore, the purity of the monomeric compounds was improved by the recrystallization method and phosphor gel column chromatography. Lastly, the crystals obtained with separation were detected as monomeric compounds using thin-layer chromatography.

### 4.3. Glucuronidation of GPC in Human Liver Microsomes

The glucuronidation incubation system consisted of Tris-HCl buffer (50 mM, pH = 7.4), magnesium chloride (4 mM), alamethicin (12.5 μg/mL), human liver microsomes (0.5 mg/mL), UDPGA (100 mM), and GPC (20 μM). The amount of organic solvent in the system did not exceed 1.5%. The incubation system was first preincubated at 37 °C for 3 min before adding UDPGA to initiate the reaction. The reaction was terminated by incubation at 37 °C for 60 min with the addition of an equal volume of iced acetonitrile (containing 30 μM PPT). After vortexing for 3 min, the supernatant obtained was centrifuged at 14,300 rpm for 15 min at 4 °C and used for LC–MS/MS analysis. Additionally, the incubation system without UDPGA was used as the negative control, and the incubation system without substrate as the blank control to demonstrate that the metabolite (GPCG) was UDPGA and microsome-dependent.

### 4.4. LC–MS/MS Analysis

The substrates and products were analyzed using an API 4000 triple quadrupole mass spectrometer. The separation was first achieved on a ZORBAX Eclipse Plus C18 column (2.1 × 100 mm, 3.5 μm) with a mobile phase of water (containing 0.1% formic acid) and acetonitrile at a flow rate of 0.5 mL/min and an injection volume of 2 μL. The gradient elution program was as follows: 0–2 min 50%B, 2–2.5 min 50%–80%B, 2.5–4 min 80%–83.5%B, 4–8 min 83.5%–86%B, 8–8.5 min 86%–50%B, and 8.5–9 min 50%B.

The mass spectrometry was performed with an electrospray ionization source (ESI), multiple reaction detection mode (MRM), and positive and negative ion mode scan detection. Parameter settings: positive ion mode, ion spray voltage is 5500 V, gas curtain gas is 25 psi, spray gas is 40 psi, auxiliary heating gas is 45 psi, the temperature is 450 °C, collision gas is 8 psi, DP voltage is 140 V (GPC), 35 V (internal standard), and CE voltage is 52 V (GPC), 8 V (internal standard), respectively. In negative ion mode, the ion spray voltage is −4500 V, DP voltage is −175.4 V (GPCG), CE voltage is −10 V (GPCG), and the rest of the parameters are the same as those in positive ion mode.

Since standards for GPC glucuronidation products are difficult to obtain, this study measured the levels of metabolites by using the peak area ratio of analyte and internal standard [37].

### 4.5. Enzyme Kinetic Analysis

The incubation volume for GPC glucuronidation was 100 μL in vitro. The product generation was first optimized with pre-experiments for microsomal protein concentration (0.1–0.7 mg/mL) (Appendix A and Appendix A) and incubation time (5–240 min) (Appendix A and Appendix A), which ensured that the product generated by liver microsomes was linear. The results show that GPCG formation was linear at a protein concentration of 0.2 mg/mL and an incubation time of 30 min. GPC stock solutions were prepared using dimethyl sulfoxide (DMSO) and the final concentration of DMSO in the incubation system was less than 1.5% (*v*/*v*). To evaluate the kinetic parameters, GPC (2–200 μM) was incubated with liver microsomes (0.2 mg/mL) for 30 min, and all other incubation conditions were as in item 4.3.

### 4.6. Chemical Inhibition Experiment

Glucuronidation metabolism of GPC in HLMs was inhibited using hecogenin, phenylbutazone, fluconazole, and magnolol. The concentrations of the four inhibitors were set to 1, 10, 50, and 200 μM, and the concentration of GPC was set to 15 μM. The incubation time and protein concentration were 0.2 mg/mL and 30 min, respectively. All the rest of the incubation conditions were as in Section 4.3. The incubation without inhibitor but with the same volume of DMSO was set to be the solvent control. The control activity was set to 100%, and the metabolic activity of the inhibitor group was compared to the control group to obtain the residual activity of the enzyme.

### 4.7. Recombinant Human UGT Enzyme Assay

The glucuronidation metabolic activity of GPC at two concentration levels (1 and 10 μM) was determined using an incubation system with 10 recombinant human UGT enzymes (UGT1A1, 1A3, 1A4, 1A6, 1A7, 1A8, 1A9, 1A10, 2B7, 2B17) (0.1 mg/mL). The incubation conditions were the same as in Section 4.3, except for the protein concentration of 0.1 mg/mL. Lastly, metabolites were detected using LC–MS/MS.

### 4.8. Species Difference Analysis

Species differences in GPC glucuronidation metabolism were examined through the liver microsomes from eight species (human: HLM, rat: RLM, mouse: DLM, dog: DLM, rabbit: RaLM, bovine: BLM, pig: PLM, monkey: MkLM). The concentrations of GPC were set at 10 and 50 μM, and the concentrations of liver microsomes from different species were 0.2 mg/mL, the incubation time was 30 min, and all the rest of the incubation conditions were as in Section 4.3.

### 4.9. Enzyme Kinetic Data Analysis

The enzyme kinetic parameters were evaluated by using GraphPad Prism 8.0 software (San Diego, CA, USA). Data were first transformed, and Eadie–Hofstee plots were plotted to facilitate the identification of kinetic models. The kinetic parameters of GPC glucuronidation in humans and different species were obtained by fitting substrate inhibition Equation (1) to the experimental data. All results are from three replicate samples of different microsomal incubations, and data are expressed as mean ± SD.
(1)$$V=\frac{V_{max}\times S\;}{K_{m}+S\times \left(1+S/K_{i}\right)}$$
where *V*, *V_max_*, *K_m,_* and *S* represent the glucuronidation rate, maximal reaction rate, Michaelis constant, and substrate concentration of GPC, respectively, and *K_i_* is the substrate inhibition constant. *CL_int_* is calculated from *V_max_*/*K_m_*.

## 5. Conclusions

GPC is one of the main components of GP. There is no report on the Phase II metabolism of GPC in vitro. In our work, the metabolites were detected via LC–MS/MS. The results showed that GPC could generate a metabolite through glucuronidation in HLMs. Using chemical inhibitors and recombinant human UGT enzymes indicated that UGT1A4 was identified as the major isoform enzyme in HLMs that catalyzes the glucuronidation metabolism of GPC. In addition, the glucuronidation of GPC in different species showed significant differences. Among them, rats exhibited the most similar metabolic rate to that of humans. Thus, rats could be used as the most suitable animal model for GP experimental research. Overall, the present study can promote the rational application of GP in the clinic, and the identification of metabolic enzymes can reduce the occurrence of interactions between GP and other drugs. The determination of the animal model can provide a model reference for subsequent experimental studies of GP.

## Figures and Tables

**Figure 1 ijms-24-01454-f001:**
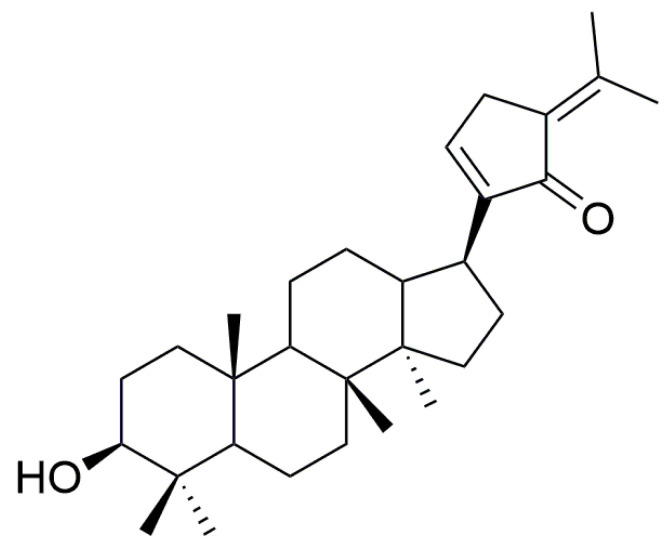
Chemical structure of GPC.

**Figure 2 ijms-24-01454-f002:**
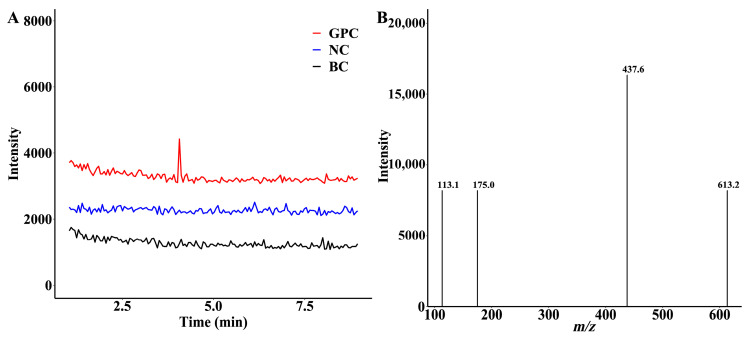
Glucuronidation metabolites of GPC in human liver microsomes. (**A**) LC–MS/MS chromatogram of the extract of the incubation system. (**B**) MS/MS spectra extracted from the chromatographic peak of GPCG at the retention time of 4.07 min.

**Figure 3 ijms-24-01454-f003:**
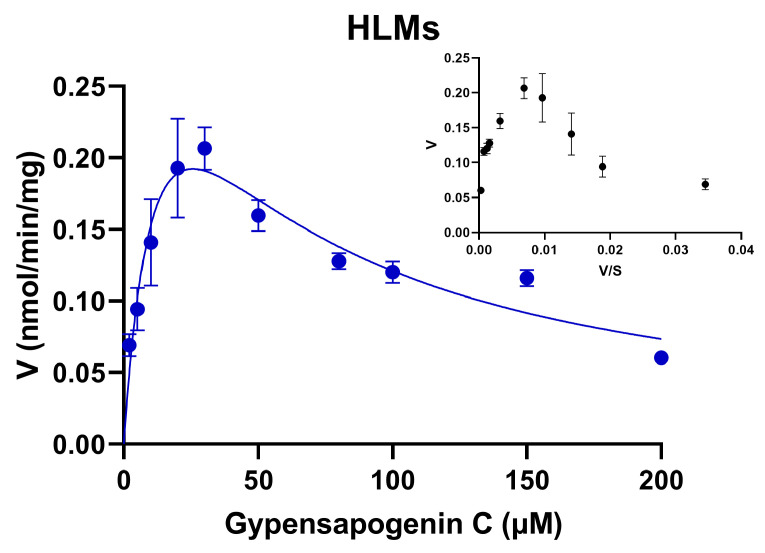
Enzyme kinetic profile of GPC in HLMs.

**Figure 4 ijms-24-01454-f004:**
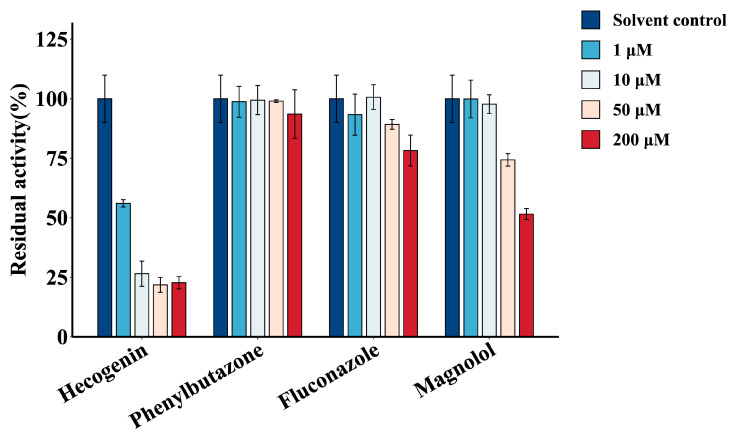
Chemical inhibition of GPC glucuronidation in human liver microsomes by four different inhibitors (1, 10, 50, and 200 μM) (hecogenin, phenylbutazone, fluconazole, magnolol) (GPC concentration of 15 μM). The incubation without inhibitor, but with the same volume of DMSO was set as the solvent control, and the control activity was set to 100%.

**Figure 5 ijms-24-01454-f005:**
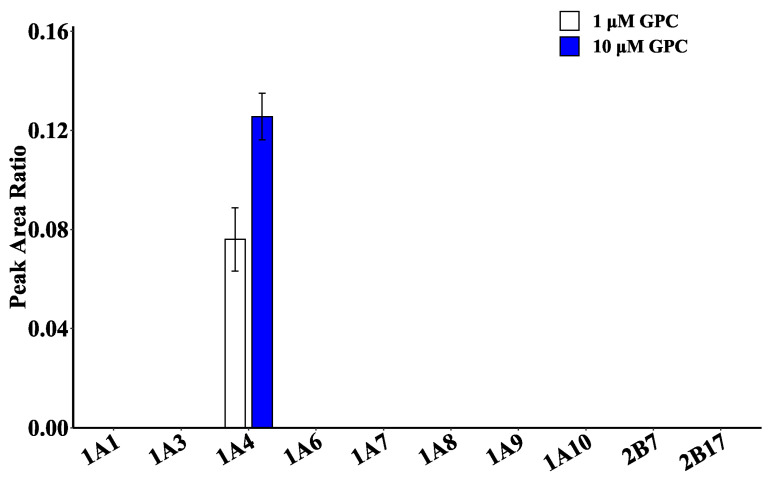
Activity assay of different recombinant human UGT enzymes (0.1 mg/mL) catalyzing glucuronidation metabolism of GPC (1, 10 μM).

**Figure 6 ijms-24-01454-f006:**
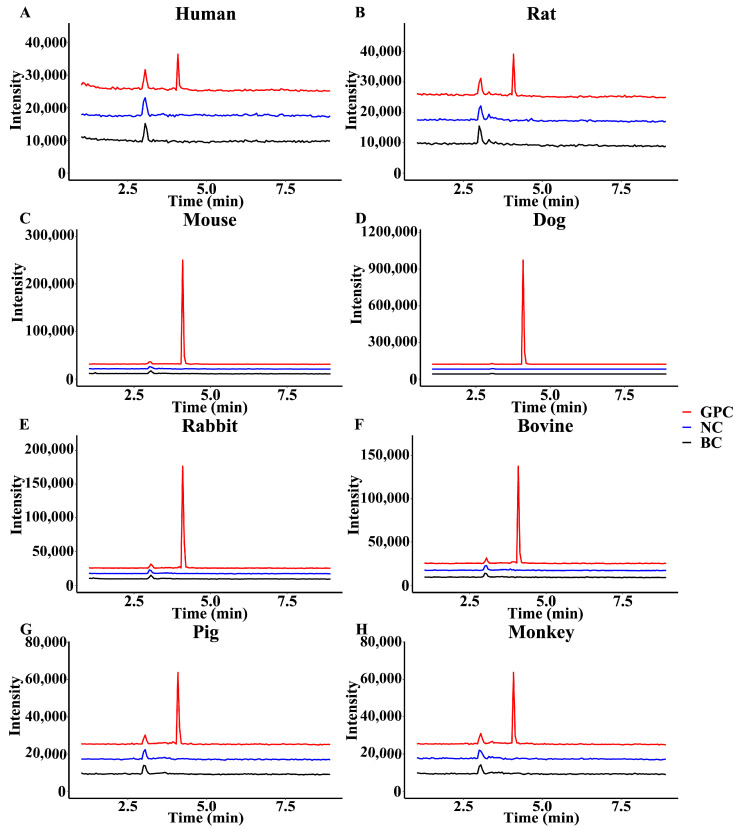
Chromatograms of GPC (10 μM) glucuronidation metabolites in (**A**) human, (**B**) rat, (**C**) mouse, (**D**) dog, (**E**) rabbit, (**F**) bovine, (**G**) pig, and (**H**) monkey liver microsomes.

**Figure 7 ijms-24-01454-f007:**
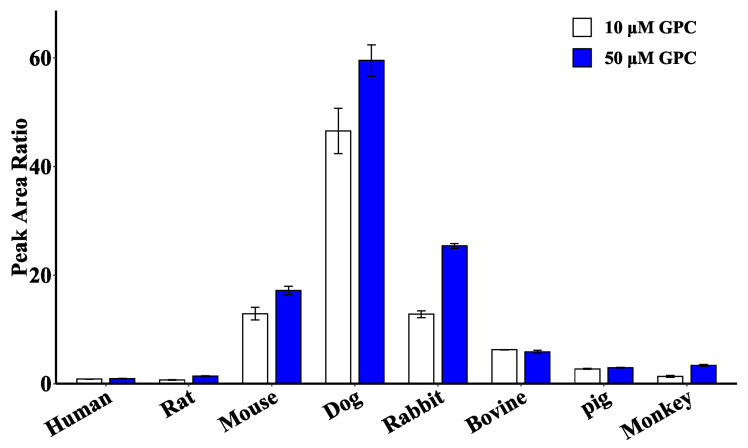
Comparison of glucuronidation metabolite production by 10 and 50 μM GPC in liver microsomes of different species.

**Figure 8 ijms-24-01454-f008:**
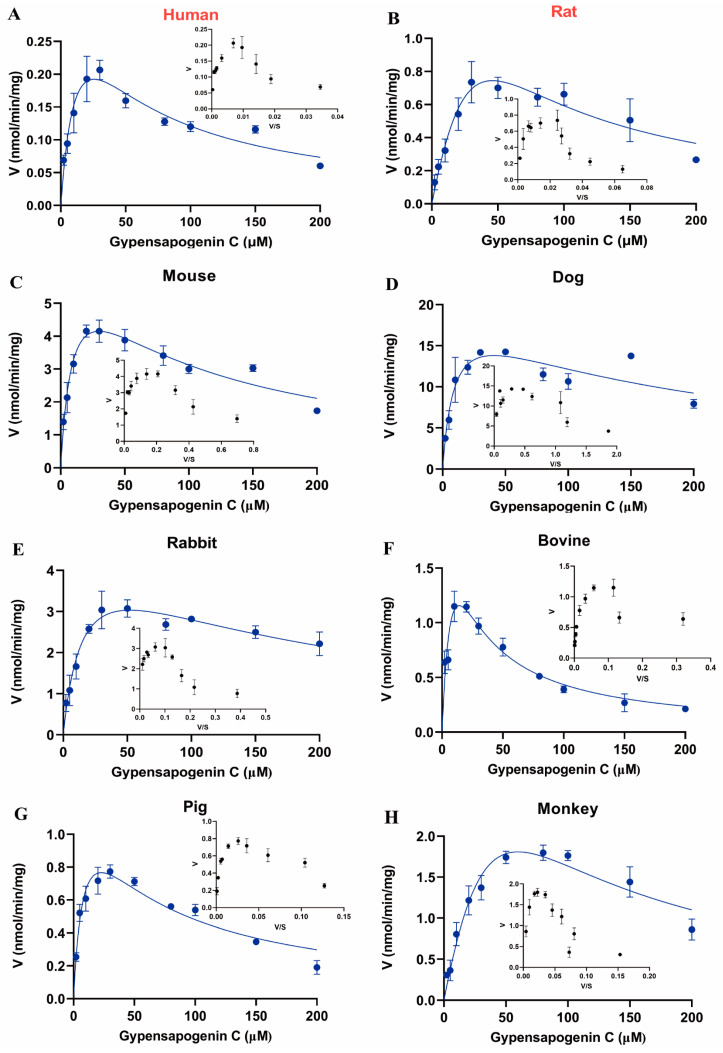
Enzymatic kinetic profiles of GPC metabolism by glucuronidation in liver microsomes of (**A**) human, (**B**) rat, (**C**) mouse, (**D**) dog, (**E**) rabbit, (**F**) bovine, (**G**) pig, and (**H**) monkey.

**Table 1 ijms-24-01454-t001:** Kinetic parameters for determination of GPC glucuronidation in HLM, RLM, MLM, DLM, RaLM, BLM, PLM, and MkLM.

Enzyme Sources	*K_m_* (μM)	*V_max_* (nmol/min/mg)	*K_i_* (μM)	*CL_int_* (μL/min/mg)
HLMs	15.36 ± 5.26	0.42 ± 0.09	42.68 ± 13.63	27
RLMs	94.07 ± 89.75	3.80 ± 3.01	22.44 ± 21.19	40
MLMs	10.04 ± 2.10	6.93 ± 0.69	89.46 ± 17.97	690
DLMs	9.56 ± 2.83	20.28 ± 2.57	173.90 ± 57.45	2121
RaLMs	16.19 ± 3.77	4.92 ± 0.57	167.50 ± 45.87	304
BLMs	12.24 ± 5.16	3.17± 0.89	16.21 ± 6.24	259
PLMs	9.58 ± 2.32	1.41 ± 0.18	54.32 ± 11.96	147
MkLMs	107.10 ± 73.18	8.29 ± 4.69	33.34 ± 23.44	77

## Data Availability

The data generated and analyzed in this study are available from the corresponding author on request.

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
