# Peer review of "Identification of Human UDP-Glucuronosyltransferase Involved in Gypensapogenin C Glucuronidation and Species Differences"

_ijms, 2023, doi:10.3390/ijms24021454_

Round 1

Reviewer 1 Report

The authors presented an investigation of the metabolism of Gypensapogenin C (GPC), the main aglycones of Gynostemma pentaphyllum, in human and other species’ hepatic microsomes. Using LC-MS/MS, in conjunction with chemical inhibitors and recombinant UGT enzymes, the authors demonstrated that UGT1A4 is a major enzyme responsible for the glucuronidation of GPC in human liver microsomes. Based on the kinetic analysis of the enzyme, the authors concluded that rat may be an appropriate animal model to evaluate GPC metabolism. Overall, this is a well-designed, innovative, and interesting paper.

The weaknesses of the paper are:

1.   Gynostemma pentaphyllum contains a variety of aglycones. More perspectives should be provided regarding why GPC was chosen in this study.

2.   In “Chemical inhibition experiment”, the hecogenin, phenylbutazone, fluconazole, and magnolol were used as chemical inhibitors, more perspectives should be provided regarding why those four inhibitors were chosen in this study.

3.   In this study, chemical inhibitors and recombinant UGT enzymes were chosen for the identification of metabolic enzymes, more perspectives should be provided regarding the reason.

4.   Writing. There were many grammar issues in the manuscript. English editing is necessary to improve the readability of the paper.

Reviewer 2 Report

Throughout the discussion the authors make several allusions to unpublished data. It would seem more appropriate if they either did not refer to these data or added them in this publication.

For the rest I think that the article is very interesting and the design, methodology, results and discussion are appropriate.

Reviewer 3 Report

Authors have carried out very pertinent study but since, animal and human studies are involved I am not sure about ethical issues. Please answer the queries raised in the manuscript, modify the title to a more suitable one, how the concentration of inhibitors have been worked out any kind of prior study on substrate concentration plot has ben created or not. There is no mention of protocol for isolation of GPC form GP please include a paragraph along with the reference. Discussion and conclusion need to be improved particularly discussion portion need to be improved and made interesting. 

Reviewer 4 Report

The authors study the Glucuronidation of GPC in the liver. They identify the enzyme involved and the chemical inhibitors. They also compare the glucuronidation process in various animal models.

I would recommend that they further explain the controls and chromatographic data, but most importantly restructure discussions to discuss and highlight their own work. Here are some major comments and some minor ones in the order they appear in the article.

P1 line 17 (P11 line 365)

I recommend separating into two sentences and explaining that the study will be done using LC-MS/MS

P1 line 22

An apology, but I don't understand why the authors say that their work “provided a reference for the rational application of GP”

Pag 3 line 83-87
Please review the sequence of presentation of the results. What are the results,.... which are the blank control or negative control (in methodology the authors indicate only one control),..... What does the analysis indicate?

Pag 3 line 97
Figure captions should explain what the figures are. For example, Figure 2A is a HPLC-MS-ESI-Q-ToF chromatograms of the extract of incubation system. Figure 2B is TOF-MS/MS spectra extracted from the chromatographic peak of GPCG at retention time of 4.07 min

Why is the GPC not visible on the chromatogram?

Pag 4 line 126 and page 11 line 335

Please check “A solvent without inhibitor but containing an equal volume of inhibitor”

Pag 4 line 129

I understand that section 2.4 is not to confirm the results of the chemical inhibitor assay, but to study the metabolic activity of recombinant human UGT enzymes. If this is correct, please rewrite the beginning of the paragraph.

Round 2

Reviewer 4 Report

I am very pleased to see the revised version of this paper, which is much improved.